# The Autophagy-Related *Musa acuminata* Protein MaATG8F Interacts with MaATG4B, Regulating Banana Disease Resistance to *Fusarium oxysporum* f. sp. *cubense* Tropical Race 4

**DOI:** 10.3390/jof10020091

**Published:** 2024-01-23

**Authors:** Huoqing Huang, Yuzhen Tian, Yile Huo, Yushan Liu, Wenlong Yang, Yuqing Li, Mengxia Zhuo, Dandan Xiang, Chunyu Li, Ganjun Yi, Siwen Liu

**Affiliations:** 1Institute of Fruit Tree Research, Guangdong Academy of Agricultural Sciences, Key Laboratory of South Subtropical Fruit Biology and Genetic Research Utilization, Ministry of Agriculture and Rural Affairs, Guangdong Provincial Key Laboratory of Tropical and Subtropical Fruit Tree Research, Guangzhou 510640, China; hqhuang07@163.com (H.H.); lele940910@163.com (Y.H.); yushanliu2020@outlook.com (Y.L.); syt226huyl@163.com (W.Y.); liyuqing_lee@163.com (Y.L.); zhuomx99@163.com (M.Z.); xiangdandan@gdaas.cn (D.X.); lichunyu@gdaas.cn (C.L.); 2Guangdong Provincial Key Laboratory of Laser Life Science, College of Biophotonics, South China Normal University, Guangzhou 510631, China; tianyzh@126.com; 3Guangdong Laboratory for Lingnan Modern Agriculture, Guangzhou 510640, China; 4Maoming Branch, Guangdong Laboratory for Lingnan Modern Agriculture, Maoming 525000, China

**Keywords:** ATG8, ATG4, autophagy, *Fusarium oxysporum* f. sp. *cubense* tropical race 4, plant disease resistance

## Abstract

Banana is one of the most important fruits in the world due to its status as a major food source for more than 400 million people. *Fusarium oxysporum* f. sp. *cubense* tropical race 4 (*Foc* TR4) causes substantial losses of banana crops every year, and molecular host resistance mechanisms are currently unknown. We here performed a genomewide analysis of the autophagy-related protein 8 (ATG8) family in a wild banana species. The banana genome was found to contain 10 *MaATG8* genes. Four *MaATG8s* formed a gene cluster in the distal part of chromosome 4. Phylogenetic analysis of ATG8 families in banana, *Arabidopsis thaliana*, citrus, rice, and ginger revealed five major phylogenetic clades shared by all of these plant species, demonstrating evolutionary conservation of the MaATG8 families. The transcriptomic analysis of plants infected with *Foc* TR4 showed that nine of the *MaATG8* genes were more highly induced in resistant cultivars than in susceptible cultivars. Finally, MaATG8F was found to interact with MaATG4B in vitro (with yeast two-hybrid assays), and MaATG8F and MaATG4B all positively regulated banana resistance to *Foc* TR4. Our study provides novel insights into the structure, distribution, evolution, and expression of the MaATG8 family in bananas. Furthermore, the discovery of interactions between MaATG8F and MaATG4B could facilitate future research of disease resistance genes for the genetic improvement of bananas.

## 1. Introduction

Banana (*Musa* spp.) is a major staple food crop with great economic importance; globally, ~124.98 million tons of bananas are produced annually [1]. Banana plants are widely distributed in tropical and subtropical regions, including Africa, Latin America, the Caribbean, Asia, and the Pacific. In some parts of Africa, bananas provide up to 25–35% of daily caloric intake to inhabitants [2]. However, banana is highly susceptible to Fusarium wilt, which is caused by *Fusarium oxysporum* f. sp. *cubense* tropical race 4 (*Foc* TR4). Fusarium wilt has a devastating economic impact on the banana industry and has gradually spread to major banana-producing countries; it was identified in Mozambique in 2013, Colombia in 2019, and Peru in 2021 (ProMusa, https://www.promusa.org/, access on 1 December 2023).

Efficient strategies to control Fusarium wilt have not yet been identified. Chemical controls are ineffective because the fungi are soil-borne and affect the vascular bundle of host plants. The repeated use of chemical fungicides also has negative impacts on the environment and human health. However, biological controls have proven unsatisfactory in field experiments [3]. Therefore, disease resistance breeding is the most promising strategy for *Foc* TR4 management. This approach will require additional research into host species’ *Foc* TR4 resistance mechanisms, particularly those involving autophagy-related proteins.

The eukaryotic autophagy pathway is a system of controlled cellular degradation that is highly conserved among animals, fungi, and plants [4,5,6]. Autophagy in plants involves a series of five steps: induction, elongation, completion, fusing, and degradation [7]. Kinase signaling activates the ATG1/ATG13 complex to form autophagosomes [8,9,10]. Autophagosomes then transport intracellular components to the vacuoles for degradation by various autophagy-related proteins at different stages of the process [11,12].

The key gene *ATG1* was the first autophagy-related gene discovered [13]. It regulates autophagy initiation and autophagosome formation [14,15]. In *Arabidopsis thaliana* and *Camellia sinensis*, at least 49 *ATG* genes have been discovered thus far [16]. Proteins involved in autophagy tend to be highly conserved across many eukaryotic species [17,18]. One such protein, ATG8, is initially processed by a cysteine protease (ATG4) to expose a glycine residue at the C-terminus [19]. Phosphatidylethanolamine (PE) is then attached to this residue by ATG7 and the E2-like enzyme ATG3. Finally, PE is removed from ATG8 by ATG4, leaving free ATG8 available to form autophagosomes [14]. Many reports have indicated that autophagosome formation fails in the absence of ATG8 [20].

Plant biotic stress responses to pathogens involve activation of numerous genes, including those encoding autophagy-related proteins [21,22,23]. Some pathogens suppress host defenses by interfering with autophagy. For example, SDE3 effector in *Candidatus Liberibacter* undermines autophagy-mediated immunity through the specific degradation of citrus ATG8 family proteins [24]. ATG8 also participates in regulating host plant disease resistance, and disruption of normal ATG8 function can therefore reduce plant defenses and promote infection. For example, the γb protein of barley stripe mosaic virus (BSMV) inhibits autophagy in the host plant *Hordeum vulgare* by directly interacting with ATG7, preventing ATG–ATG8 interactions and thus promoting successful infection [25]. In apple (*Malus domestica*), MdATG8i decreases pathogen sensitivity by interacting with the target protein MdEF-Tu. The *Valsa Mali* effector Vm1G-1794 inhibits autophagy by competitively binding to MdATG8i, thus weakening plant resistance. In contrast, *MdATG8i* overexpression significantly improves pathogen resistance [21]. Prior studies suggested that the causative agent of apple rot manipulates the apple autophagy pathway through secretion of specific effector proteins. ATG4 and ATG6/Beclin-1 were also shown to participate in autophagy induced by biotic and abiotic stressors [26]. Such studies demonstrated the important roles of ATG8s in host plant infection resistance. Previous reports only simply identified banana ATG genes at the genome scale or characterized their roles in *Foc* TR4 resistance [27]. However, the molecular-level differences between resistant and susceptible cultivars are still unknown.

In the present study, pathogen-induced *ATG8* family members were identified throughout the banana genome. Gene structures, conserved domains, phylogenetic relationships, chromosomal locations, and expression patterns of *MaATG8* genes in response to *Foc* TR4 infection were analyzed. Furthermore, MaATG8 functions were established through the assessment of interactions with other ATGs and the effects of gene silencing in banana during *Foc* TR4 infection. This study provides valuable new insights that will inform future research into the molecular mechanisms of autophagy and resistance breeding in banana.

## 2. Materials and Methods

### 2.1. Culture Conditions for Plants and Fungi

The Cavendish banana (*Musa* spp. AAA group) cultivars ‘ZhongJiao No. 6’ (ZJ6) and ‘Brazilian’ (BX) were used as resistant cultivar and susceptible cultivar in this research. ZJ6 (from Fruit Research Institute, Guangdong Academy of Agricultural Sciences, China) is artificially bred from BX. The banana plantlets at the five- to six-leaf stage were used in a controlled greenhouse at 28 °C under a 14/10 h light/dark cycle. *Nicotiana benthamiana* plants, a plant model with a clear genetic background, were grown for ~28 d in a greenhouse at 24 °C under a 16/8 h light/dark cycle. Prior to plant inoculation, *Foc* TR4 strain II5 (NRRL#54006), a model strain with a clear genetic background in research studies of the banana host and *Foc*, was cultured on potato dextrose agar (PDA) for 6 d or in potato dextrose broth (PDB) for 3 d.

### 2.2. Identification of ATG8 Genes in Multiple Species

The amino acid (aa), genomic DNA (gDNA), and coding sequences (CDS) of *Musa. acuminata* were downloaded from the Banana Genome Database (http://banana-genome-hub.southgreen.fr/, access on 7 October 2023). A hidden Markov model (HMM) for ATG8 (PF02991) was downloaded from the Pfam database (http://pfam.xfam.org/, access on 7 October 2023). The *M. acuminata* genome was then searched using PF02991 as a query with the Simple HMM Search in TBtools. NCBI CDD search (https://www.ncbi.nlm. nih.gov/cdd, access on 7 October 2023) was used for domain validation in the resulting sequences. Using candidate protein IDs, the aa, gDNA, and CDS were obtained with the TBtools sequence extraction tool (Fasta Extract) [28].

### 2.3. Chromosomal Locations of MaATG8s

Chromosome location mapping was conducted with the Gene Location Visualize from GTF/GFF function of TBtools based on the *MaATG8* gene IDs from the *M*. *acuminata* gff3 files. *M*. *acuminata* genome sequences were analyzed in pairwise comparisons using One Step MCScanX in TBtools. *MaATG8* gene family collinearity was visualized in other species with Multiple Synteny Plot in TBtools. The chromosomal coordinates of *MaATG8* genes were extracted from the annotated *M*. *acuminata* genome files and visualized using TBtools.

### 2.4. RNA Extraction and Quantitative Reverse Transcription (qRT)-PCR

RNA from banana roots was extracted with an RNA extraction kit (Accurate Biotech, Guluo, Hunan, China) following the manufacturer’s instructions. Briefly, the banana roots were ground into powder in liquid nitrogen at low temperatures. The RNA extraction kit was used to extract the RNA from roots. The kit contained RNase-free Recombinant DNase I to eliminate genomic DNA. Reverse transcription was performed with the Evo M-MLV One Step RT-PCR Kit (Accurate Biotech, Hunan, China). qRT-PCR was performed on a StepOne real-time PCR system (Applied Biosystems, Waltham, MA, USA) with ChamQ Universal SYBR qPCR Master Mix (Vazyme Biotech, Nanjing, China) following the manufacturer’s instructions. Gene relative expression levels were determined using the 2^−∆∆Ct^ method with the endogenous reference gene *MaTUB* (β-tubulin). There were three biological replicates of each sample type, and qRT-PCR was performed in technical triplicate. All primer pairs are shown in Appendix A.

### 2.5. Banana Transcriptomic Analysis

The roots of BX and ZJ6 plants were inoculated with *Foc* TR4 strain II5 and collected after 18, 32, and 56 h. There were three biological replicates per cultivar for each time point. The Illumina HiSeq×10 platform was used to generate 150 bp paired-end reads with Gene Denovo Biotechnology Co. (Guangzhou, China). Filtered reads were aligned to the *M*. *acuminata* genome using HISAT2 v2.0.5 [29]. Differential gene expression was calculated using the ‘DESeq2’ R package. Genes were considered significantly differentially expressed at|log2 (fold change)| > 1 (*p* < 0.05), indicating genes were upregulated or downregulated.

### 2.6. Plasmid Construction

*MaATG8* genes were cloned from cDNA generated from Cavendish banana roots following the manufacturer’s instructions (2 × Phanta Max Master Mix, Vazyme Biotech, Nanjing, China). The amplified fragments were ligated into the BamHI-digested pCAMBIA1300-GFP (Green Fluorescent Protein, GFP) empty vector (MiaoLingBio, Wuhan, China), or SmaI-digested pGBKT7 or pGADT7 empty vector (Clontech, Mountain View, CA, USA) using the In-Fusion Cloning Kit (Vazyme Biotech, Nanjing, China). The primers used for plasmid construction are listed in Appendix A. Individual colonies containing each construct were verified via PCR and sequencing (Sangon, Shanghai, China).

### 2.7. Subcellular Localization

The *MaATG8F* CDS was cloned into the pCAMBIA1300-35S:GFP empty vector, and the resulting constructs were transformed into *Agrobacterium tumefaciens* strain GV3101 (WeidiBio, Shanghai, China). *Agrobacterium*-mediated transient expression in *N*. *benthamiana* leaves was performed as previously described with some modifications [30]. Briefly, transformed cells were cultured for 8 h in liquid Luria–Bertani (LB) medium at 28 °C. Bacterial cells were harvested via centrifugation, then resuspended in infiltration medium containing 10 mM 2-(4-Morpholino) ethanesulfonic acid, 10 mM MgCl_2_, and 200 μM acetosyringone (pH = 5.6) (Sangon Biotech, Shanghai, China). The concentration of the cell suspension was adjusted to an OD_600_ of 1.0, and then the suspension was incubated for 2–3 h at room temperature prior to infiltration into the leaves of four-week-old *N*. *benthamiana* plants. The leaves were observed under a confocal microscope (LSM 710, Carl Zeiss, Oberkochen, Germany) at 72 h postinfiltration.

### 2.8. Inoculation Assays

To determine banana plant phenotypes after *Foc* TR4 infection, banana cultivars of two different resistance levels were selected: BX and ZJ6. ZJ6 is a resistant cultivar whereas BX is highly susceptible to *Foc* TR4. The inoculation assays were performed as previously described with some modifications [31]. Briefly, these two banana cultivars were grown in a greenhouse to the five- to six-leaf stage for use in inoculation assays. Banana cultivars were dipped with *Foc* TR4 suspension at a concentration of 1×10^7^ conidia/L for half an hour. The plants were then replanted in soil. A total of 30 banana plantlets were used for each treatment. Each experiment was repeated three times independently. Banana corms were observed and photographed at 35 d postinoculation and graded for disease appearance as follows: 0 (no symptoms), 1 (some brown spots in the inner rhizome), 2 (less than 25% of the inner rhizome browning), 3 (up to 75% of the inner rhizome browning), and 4 (entire inner rhizome and pseudostem dark brown and dead). The disease index was calculated as follows:100 × [∑ (total number of plants × disease grade)/(total number of plants × 4)]

### 2.9. Yeast Two-Hybrid (Y2H) Assays

The Y2H system was used to identify protein–protein interactions in vitro. The *MaATG4B* CDS was cloned inframe into the bait vector pGBKT7, and the CDSs of *MaATG8* genes were cloned inframe into the prey vector pGADT7 using the In-Fusion Cloning Kit (Vazyme Biotech, Nanjing, China). The primers used for plasmid construction are listed in Appendix A. The Y2H Gold yeast strain (WeidiBio, Shanghai, China) was cotransformed with the resulting bait and prey vectors to identify positive interactions following the manufacturer’s protocol (Clontech, Mountain View, CA, USA). The cotransformants of pGBKT7-53 + pGADT7-T and pGBKT7-lam + pGADT7-T are presented as positive control and negative control, respectively. SD/-leucine-tryptophan-adenine-histidine medium (SD/-LWAH) with the addition of X-α-gal was used to screen the positive interactors at 30 °C in the dark for about 3 days, and SD/-leucine-tryptophan medium (SD/-LWAH) was used to screen the positive transformants. X-α-gal was used to confirm the positive interactions by turning the colonies blue. The analysis of this experiment was to observe if the yeast colonies grew on SD/-LWAH and SD/-LW media, and if colonies turned blue on the SD/-LWAH medium with added x-α-gal. If the colonies grew on SD/-LW and appeared blue on SD/-LWAH after about 3 days, the results would indicate that they were interacting. The plates were then photographed using a Nikon camera.

### 2.10. Gene Silencing

dsRNAs of ~300 bp in size that were complementary to *MaATG8F* and *MaATG4B* were generated with the T7 RNAi Transcription Kit following the manufacturer’s protocol (Vazyme Biotech, Nanjing, China). The PCR technique was used to add the T7 promoter sequence to both ends of the RNA interference (RNAi) target fragments. The primers used for PCR are listed in Appendix A. Sequences including the T7 promoters were purified and used as the templates for dsRNA amplification. Infection assays were then performed to establish the effects of knocking out target genes performed as in a previous study with some modifications [32]. Specifically, banana leaves of the same age were detached and pressed with the tip of a pipette head to make even wounds, then treated with 10 μL dsRNA at a concentration of 500 ng/μL with 0.02% Silwet L-77 (Solarbio Science & Technology, Beijing, China). The sites in the same leaves treated with 10 μL water with 0.02% Silwet L-77 were used as a control. The Silwet L-77 was a surfactant used for facilitating the adsorption and spread of dsRNA or water on the banana leaves. The dsRNA solution and water were allowed to dry for about 1 h, then mycelial plugs of 5 mm in diameter were taken from plates containing 6-day-old *Foc* TR4 strain II5. Plugs were taken only from the growing edges and were placed onto the portion of banana leaves treated with dsRNA. The mycelium was facing down, touching the banana leaves. A total of 10 banana leaves were used for each treatment. Each experiment was repeated three times independently. All leaves were then moved to plastic trays, each of which was lined with two layers of moistened paper towel. To maintain sufficiently high humidity, each tray was covered with plastic film. Trays containing inoculated leaves were incubated at 28 °C in the dark for 10 days. Leaves were then photographed, and the fungal lesion size was quantified in ImageJ V1.8.0 software (https://imagej.nih.gov/ij/, access on 20 November 2023). Leaves were also harvested for qRT-PCR to verify gene silencing.

### 2.11. Trypan Blue Staining

Trypan blue staining was performed as previously described [33] with some modifications. Briefly, banana leaves were harvested and soaked in trypan blue solution (0.02% trypan blue in 1:1:1:1:8 phenol: glycerol: lactic acid: water: ethanol (*v*/*v*/*v*/*v*/*v*)) at room temperature overnight. Leaf samples were destained in 75% alcohol several times until the destaining solution remained clear. The samples were then photographed using a Nikon camera.

## 3. Results

### 3.1. Genomewide Identification of MaATG8 Genes

Analysis of the *M*. *acuminata* genome revealed 10 putative MaATG8 genes (Appendix A). The longest MaATG8 protein (MaATG8D) was 141 amino acids (aa) in length, and the shortest (MaATG8E) was 80 aa. A neighbor-joining phylogenetic tree constructed from ATG8 genes in banana, *A*. thaliana, rice, citrus, and ginger showed five major phylogenetic clades, which were conserved in banana (10 genes), *A*. *thaliana* (8 genes), rice (6 genes), citrus (11 genes), and ginger (20 genes) (Appendix A, Figure 1A). MaATG8E alone formed a separate branch (Figure 1A). Overall, this phylogenetic analysis showed that ATG8 homologs were conserved among several plant species of varying evolutionary distances from one another. Furthermore, aa sequence alignment showed that the MaATG8 proteins were highly evolutionarily conserved (Figure 1B).

### 3.2. MaATG8 Gene Structures and Conserved Motif Analysis

Gene structure analysis showed notable differences in the numbers and lengths of exons in *MaATG8* genes. There were nine motifs identified across the *MaATG8* genes (Figure 2A). *MaATG8C*, *MaATG8A*, *MaATG8F*, *MaATG8J*, *MaATG8E*, and *MaATG8G* contained just two motifs each (motifs 1 and 2). *MaATG8H* had three motifs (2, 6, and 8); *MaATG8B* contained four motifs (1, 2, 3, and 5); and *MaATG8I* had five motifs (1, 2, 3, 5, and 9). *MaATG8D* contained the largest number of motifs at seven (motifs 2, 6, and 9 and two each of motifs 4 and 7) (Figure 2A). *MaATG8D* also contained two exons and no untranslated region (UTR), whereas the other nine MaATG8 genes had five exons each (Figure 2B) and one (*MaATG8B* and *MaATG8I*), two (*MaATG8G*, *MaATG8J*, and *MaATG8H*), or three (*MaATG8E*, *MaATG8C*, *MaATG8A*, and *MaATG8F*) UTRs (Figure 2B). These results indicate varying degrees of evolutionary conservation and divergence between the *MaATG8* genes.

### 3.3. Predicted Cis-Acting Elements in MaATG8 Promoters

To understand the putative functional properties of *MaATG8* gene family members, the PlantCARE database was used to predict cis-acting elements in the promoter regions (classified as 2000 bp upstream of the transcription start site for each gene). This analysis revealed many cis-acting elements, including low-temperature responsiveness, defense and stress responsiveness, and hormone responsiveness elements in addition to MYB binding sites. Six *MaATG8s* had abscisic acid (ABA) response elements and six had auxin responsiveness elements (Figure 3). For example, *MaATG8A* contained multiple ABRE (ABA-responsive) elements. Five *MaATG8s* were predicted to respond to methyl jasmonate (MeJA) (Figure 3); *MaATG8D*, *MaATG8E*, and *MaATG8J* each contained multiple binding sites for MeJA responsiveness. Four *MaATG8s* were predicted to be salicylic acid (SA)-responsive. The *MaATG8s* also included elements for stress responses (Figure 3). These results suggest that *ATG8* genes play important roles in banana stress and hormone responses.

### 3.4. Chromosomal Distribution of MaATG8s

Analysis of the 10 *MaATG8* locations showed that they were distributed across chromosome (Chr) 2, Chr 4, Chr 5, Chr 8, Chr 10, and Chr 11. The other banana chromosomes did not contain any *ATG8* genes (Figure 4). Chr4 contained more MaATG genes (four) than any other chromosome: *MaATG8B*, *MaATG8C*, *MaATG8D*, and MaATG8E. Chr5 contained two *MaATG8* genes (*MaATG8H* and *MaATG8F*); the remaining *MaATG8s* were present on Chr2, Chr8, Chr10, and Chr11, which contained just one *MaATG8* each (*MaATG8A*, *MaATG8G*, *MaATG8I*, and *MaATG8J*, respectively). These results indicate that *MaATG8s* are distributed unevenly throughout the genome.

### 3.5. Involvement of MaATG8 Family Members in the Foc TR4 Infection Response

The resistant *M*. *acuminata* cultivar ZJ6 and the susceptible cultivar BX were inoculated with *Foc* TR4 strain II5 at the five- to six-leaf stage. At 35 d after inoculation, the corms of BX plants showed significant browning symptoms; the disease incidence exceeded 95%, and the disease index was higher than 78% (Figure 5A,B). Among ZJ6 plants, only 43.3% showed mild disease symptoms (Figure 5A,B), and the disease index in ZJ6 plants reached only 18.3%, significantly lower than that of BX (Figure 5C). This demonstrated ZJ6 resistance to *Foc* TR4. To investigate *MaATG8* gene expression profiles during the early infection process (at 18, 32, and 56 h postinoculation), we performed transcriptomic analyses of the susceptible and resistance cultivars. Nine of the *MaATG8* genes were induced in both the resistant and susceptible cultivars; only MaATG8I was not induced (Appendix A; Figure 5D). Furthermore, the remaining nine genes were all highly expressed in the resistant cultivar, but only *MaATG8C* (*Macma4_04_g32300*), *MaATG8G* (*Macma4_08_g10850*), *MaATG8A* (*Macma4_02_g16130*), *MaATG8D* (*Macma4_04_g36560*), and *MaATG8H* (*Macma4_05_g30830*) were continuously induced during the infection process (Appendix A; Figure 5D). *MaATG8F* (Macma4_05_g02860), *MaATG8J* (*Macma4_11_g04160*), and *MaATG8E* (*Macma4_04_g42290*) were most strongly induced in the resistant cultivar (Appendix A; Figure 5D). These results showed that MaATG8 genes were responsive to *Foc* TR4 infection and were more highly expressed in the roots of the resistant cultivar (ZJ6) than in the susceptible cultivar (BX), clearly demonstrating participation of *MaATG8* genes in the banana immune response.

### 3.6. MaATG8F Silencing Reduced Foc TR4 Resistance

To investigate the biological function of *MaATG8F* in the *Foc* TR4 infection response, wounded banana leaves were treated with *MaATG8F*-dsRNA or a water control, then plugs of *Foc* TR4 II5 were placed on the same areas. As expected, *MaATG8F* expression was strongly downregulated after treatment with *MaATG8F*-dsRNA compared with that in the water-treated control leaves, demonstrating successful *MaATG8F* silencing (Figure 6A). Furthermore, leaves treated with *MaATG8F*-dsRNA showed more extensive necrosis than water-treated control leaves, as confirmed with both a visual comparison of the leaves (Figure 6B,C) and trypan blue staining (Figure 6D). To determine MaATG8F subcellular localization, a MaATG8F–green fluorescent protein (GFP) construct was generated with the pCAMBIA1300-35S:GFP vector, which was transiently expressed in *N*. *benthamiana* leaves. Microscopic observations of *N*. *benthamiana* leaves collected at 3 d post-agro-infiltration revealed nuclear and cytoplasmic localization of GFP:MaATG8, identical to the results of leaves infiltrated with the GFP empty vector (Figure 6E). Overall, these results indicated that MaATG8F positively regulated banana disease resistance to *Foc* TR4.

### 3.7. MaATG4B Interacted with MaATG8F

To investigate the molecular mechanism underlying MaATG8-mediated activation of the host immune response, yeast two-hybrid (Y2H) assays were performed to identify potential host targets of MaATG4, which are autophagy-related members of the ATG family. Interactions of ATG4s and ATG8s were previously identified in *Arabidopsis* [34]. *MaATG4A* and *MaATG4B* were found to be induced during *Foc* TR4 infection, whereas *MaATG4C* was not (Figure 7A). Notably, *MaATG4B* was only induced in the resistant cultivar ZJ6, whereas *MaATG4A* was induced in both ZJ6 and BX (Figure 7A). Thus, MaATG4B was selected for confirmation of physical interactions with MaATG8 homologs. The full-length coding sequences (CDSs) of *MaATG4B* and *MaATG8* genes were cloned into the bait and prey vectors, respectively. Colony growth of cotransformants on selection medium containing X-α-gal confirmed pairwise interactions between MaATG4B and all 10 MaATG8 proteins in yeast. Only MaATG8F interacted with MaATG4B (Figure 7B). These results indicate that the interactions of MaATG8F and MaATG4B could also happen in banana plants.

To investigate the biological function of MaATG4B in response to *Foc* TR4 infection, banana leaves were treated with MaATG4B-dsRNA or a water control. qRT-PCR analysis showed significant MaATG4B downregulation in dsRNA-treated leaves compared with that in water-treated control leaves (Figure 7C), indicating that dsRNA treatment silenced MaATG4B as expected. To confirm whether silencing *MaATG4B* influenced *Foc* TR4 resistance, dsRNA-treated leaves were inoculated with *Foc* TR4 plugs. The dsRNA-treated leaves showed larger necrotic areas than the controls, as confirmed with both a visual comparison of the leaves (Figure 7D,E) and trypan blue staining (Figure 7F). Taken together, these results indicate that MaATG4B functions as a positive regulator of banana defense responses and interacts with the autophagy-related protein MaATG8F.

## 4. Discussion

Plant growth is a continuous process that is regulated by several genetic and environmental factors. Stressors such as salt, high temperature, drought, and pathogen infection in particular have substantial influences on plant growth and yield throughout the world. Plants have evolved various mechanisms to protect themselves from environmental stressors, enabling survival. Autophagy is one of the most important strategies that allows plants to survive under adverse conditions [18,35,36]. The molecular mechanisms by which members of the autophagy-related ATG8 protein family function in banana have not previously been characterized. In the present study, 10 *MaATG8* genes were identified throughout the banana genome. Phylogenetic analysis of *ATG8* genes in *A*. *thaliana*, rice, citrus, ginger, and banana demonstrated relatively high conservation between species, indicating that ATG8 proteins may perform similar functions in different species.

*MaATG8* was here found to be induced by *Foc* TR4 infection and was more highly expressed in a resistant than a susceptible banana cultivar. This led to the hypothesis that *MaATG8* expression levels in resistant cultivars may be associated with their mechanisms of disease resistance. Silencing *MaATG8* in banana leaves enhanced host susceptibility to *Foc* TR4 infection, which might be one of explanations for the hypothesis. An increasing number of studies have shown that single nucleotide polymorphisms (SNPs) play important roles in various plant processes [37,38,39]. A cis-acting element analysis here revealed that the MaATG8 promoter sequences contained a large number of resistance-related cis-elements. This suggested that the *MaATG8* promoter elements of resistant and susceptible cultivars contained SNPs, leading to differences in the binding abilities of associated transcription factors, resulting in cultivar-specific differences in *MaATG8* gene expression. This is another possible explanation for differential *MaATG8* expression between resistant and susceptible cultivars. Due to the difficulty of banana plant transformation and the length of time required, *MaATG8* transgenic plants have not yet been obtained. To verify the biological functions of *MaATG8s*, banana suspension cells should be transformed to generate transgenic plants, including those overexpressing *MaATG8s* or expressing CRISPR/Cas9-edited versions of *MaATG8s*. Further studies should also address the mechanisms by which *MaATG8s* induce autophagy to degrade effectors secreted by *Foc* TR4. These mechanisms may protect banana plants from pathogen infection.

Autophagy-related proteins are known to play important roles in plant life processes, including pathogen resistance. For example, MdATG8i can mediate disease resistance and drought tolerance in apples [21,40]. Many mechanisms of interaction between ATG4 and ATG8 have been revealed previously. For example, the cysteine protease ATG4 facilitates normal ATG8 functioning by first hydrolyzing the C-terminal arginine residue, exposing a glycine residue for PE binding, then by cleaving ATG8 from PE to enable autophagosome formation [14,19]. The results of the present study verified interactions between MaATG4 and MaATG8 in vitro, suggesting that MaATG4–MaATG8 interactions also regulated autophagy in banana. Interestingly, Y2H assays showed that MaATG4 could not interact with every one of the MaATG8 family members, indicating that some of these proteins may have evolved new functions in banana. Further studies should be conducted to determine if the other two MaATG4 homologs in *M*. *acuminata* can interact with all MaATG8 family members.

ATG6 is another protein with an important role in ATG8-mediated autophagosome formation. In rice, the C-terminus of a *Rhabdovirus* glycoprotein interacts with SnRK1B, promoting ATG6b phosphorylation. Rice ATG6b can also interact with the *N*-terminus of the viral glycoprotein; this connects the glycoprotein with ATG8 on the autophagosome membrane, promoting removal of the glycoprotein into the autophagosome for degradation [22]. A prior study indicated that ATG6, ATG8, and ATG4 all interact to form a protein complex. However, whether this is true in banana, and whether such a complex participates in *Foc* TR4 resistance, remains unknown. Previous research demonstrated that interactions between ATG8 and ATG8-family interacting motif (AIM)-containing proteins participate in cell-selective autophagy and regulate plant disease resistance [41,42]. Our research group determined with Y2H assays that an AIM-containing protein interacts with MaATG8 (unpublished data), but the relationship between this interaction and disease resistance requires further study.

In summary, a total of 10 *MaATG8* genes were here identified in the banana genome. Phylogenetic relationships, gene structures, chromosomal locations, and expression pattern analyses were conducted for all 10 *MaATG8* genes. Nearly all of the *MaATG8* genes were upregulated in disease-resistant cultivars after *Foc* TR4 infection. Based on the results of *MaATG8* expression analyses, the role of *MaATG8* was explored further. *MaATG8F* silencing indicated that this gene positively regulated banana resistance to *Foc* TR4. Y2H results showed that MaATG4B could interact with MaATG8F in vitro but not with the other MaATG8 homologs. This study provides novel resistant gene resources for subsequent functional research of autophagy-related proteins in banana.

## 5. Conclusions

In the present study, 10 *MaATG8* genes were identified in the banana genome, some of which were determined to be involved in the host immune response against *Foc* TR4. Nine of the ten *MaATG8* genes were significantly induced during the infection process in the resistant and susceptible cultivars, with the exception of *MaATG8I*. The results of several analyses indicated that MaATG8F partially regulated banana disease resistance to *Foc* TR4. *MaATG4A* and *MaATG4B* were strongly upregulated in response to *Foc* TR4 infection, with *MaATG4B* only induced in the disease-resistant banana cultivar. MaATG4B was found to interact with MaATG8F (but not with other MaATG8s) and to partially regulate banana disease resistance to *Foc* TR4. Taken together, this comprehensive analysis of the MaATG8 family and interaction between MaATG8F and MaATG4B in a wild banana species provides valuable new information to facilitate mining of related resistance genes for future genetic improvement of banana cultivars.

## Figures and Tables

**Figure 1 jof-10-00091-f001:**
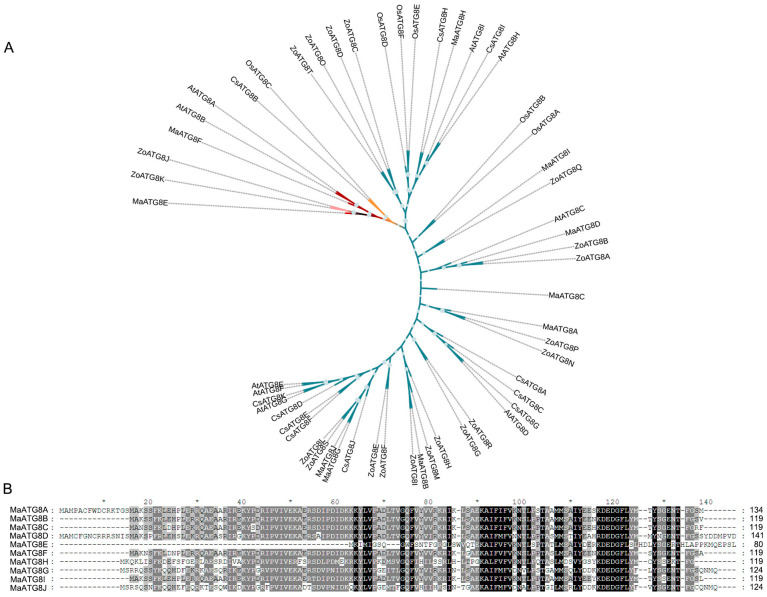
Genomewide identification of *ATG8* genes. (**A**) Phylogenetic analysis of MaATG8 proteins in banana, *Arabidopsis thaliana*, citrus, rice, and ginger. The subgroups are indicated by different frame colors. (**B**) Amino acid sequences alignment of 10 MaATG8 proteins in banana.

**Figure 2 jof-10-00091-f002:**
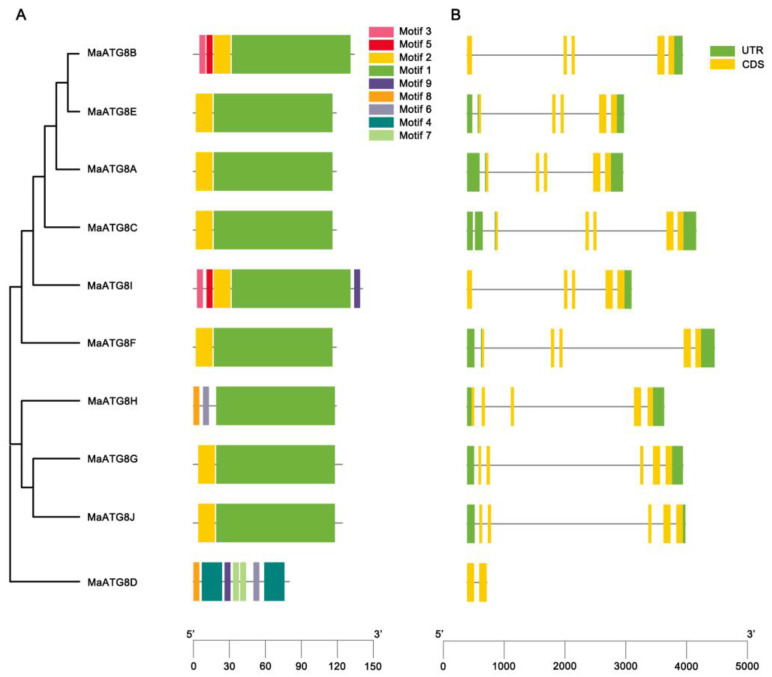
Genomewide identification of banana *ATG8* genes. (**A**) Phylogenetic analysis and motif analysis of MaATG8 proteins in banana, *A*. *thaliana*, citrus, rice, and ginger. The subgroups are indicated by different frame colors. (**B**) Gene structures of MaATG8 genes in banana. Yellow boxes indicate exons (coding sequence), green boxes indicate untranslated regions and gray lines indicate introns.

**Figure 3 jof-10-00091-f003:**
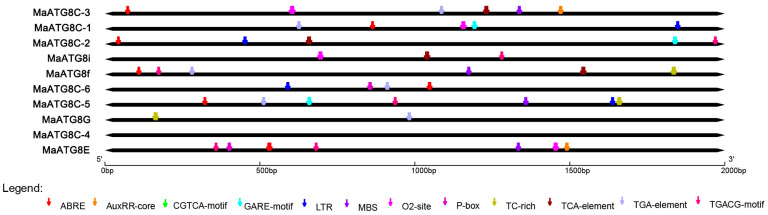
The analysis of cis-acting elements in the *MaATG8* gene promoters.

**Figure 4 jof-10-00091-f004:**
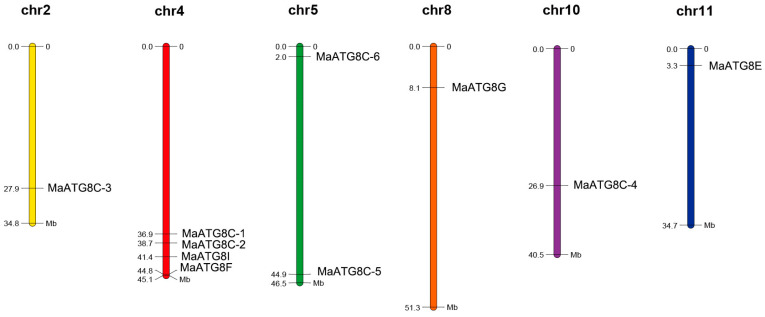
Chromosomal distribution of banana *MaATG8* genes. The chromosome number is shown above each chromosome. The left number represents the length of chromosomes.

**Figure 5 jof-10-00091-f005:**
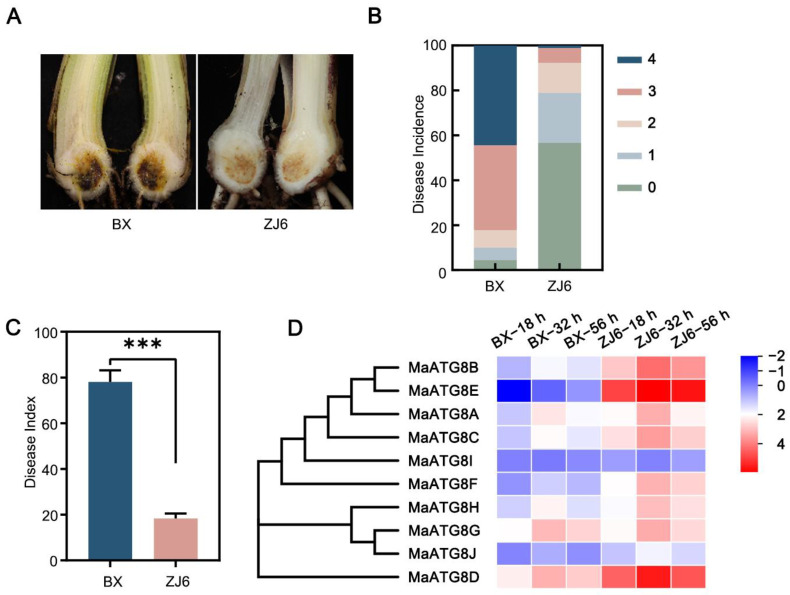
*MaATG8* genes were induced during *Foc* TR4 infection. (**A**) Disease symptoms, (**B**) disease incidence, and (**C**) disease index in resistant and susceptible plants inoculated with II5 strains for 5 weeks. ZJ6 is resistant cultivar while BX is susceptible cultivar. Data are presented as means ± SDs (*n* = 3). “***” symbol atop the columns indicates a significant difference at *p* < 0.001 (two-tailed Student’s *t*-test). (**D**) Heatmap for the *MaATG8* genes in resistant and susceptible plants inoculated with II5 strains for different time points (18 h, 32 h, 56 h).

**Figure 6 jof-10-00091-f006:**
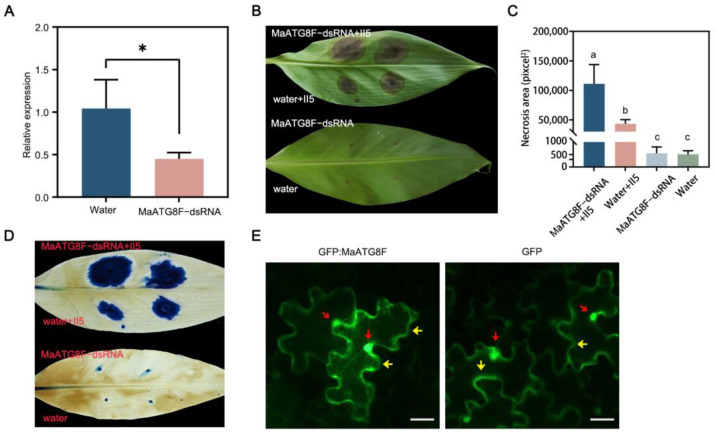
MaATG8F positively regulated banana disease resistance against *Foc* TR4. (**A**) The expression level of *MaATG8* in dsRNA- and water-treated banana leaves by qRT-PCR analysis. Data are presented as means ± SDs (*n* = 3). “*” symbol atop the columns indicates a significant difference at *p* < 0.05 (two-tailed Student’s *t*-test). (**B**) The disease symptoms and (**C**) necrosis area of dsRNA- and water-treated banana leaves. The pictures were taken 10 days postinoculation. The necrosis area was calculated with ImageJ V1.8.0 software. Data are presented as means ± SDs (*n* = 3). Different letters atop the columns indicate a significant difference at *p* < 0.01 (one-way analysis of variance). (**D**) Trypan blue staining of banana leaves. (**E**) Subcellular localization of MaATG8F in *N*. *benthmiana* leaves. Photographs were taken at 3 days postinfiltration. The empty vector (GFP) was used as control. Red arrowheads indicate nucleus, and yellow arrowheads indicate cytoplasm. Scale bar = 10 μm.

**Figure 7 jof-10-00091-f007:**
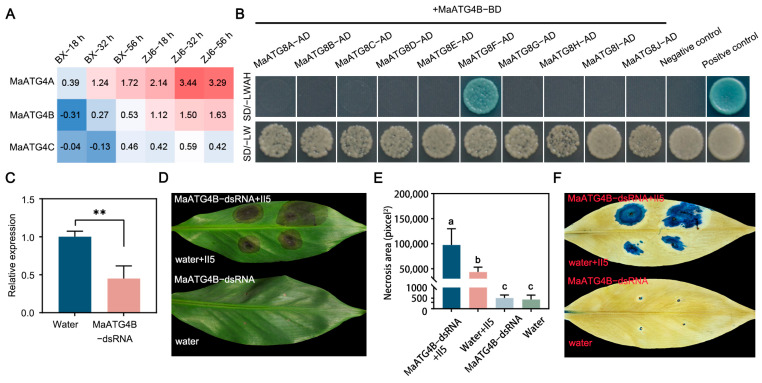
MaATG4B positively regulates the banana resistance against *Foc* TR4. (**A**) Heatmap for the *MaATG4* genes in resistant and susceptible plants inoculated with II5 strains at different time points (18 h, 32 h, 56 h). (**B**) MaATG4B interacts with MaATG8F. The cotransformants of p53+pGADT7-T and lam+pGADT7-T are presented as positive control and negative control, respectively. SD/-LWAH is presented as SD/-leucine-tryptophan-adenine-histidine, SD/-LW is presented as SD/-leucine-tryptophan. (**C**) The expression level of MaATG4B in dsRNA- and water-treated banana leaves from qRT-PCR. Data are presented as means ± SDs (*n* = 3). “**” symbol atop the columns indicates a significant difference at *p* < 0.01 (two-tailed Student’s *t*-test). (**D**) The disease symptoms and (**E**) necrosis area of dsRNA- and water-treated banana leaves. The pictures were taken 10 days postinoculation. The necrosis area was calculated with ImageJ V1.8.0 software. Data are presented as means ± SDs (*n* = 3). Different letters atop the columns indicate a significant difference at *p* < 0.01 (one-way analysis of variance). (**F**) Trypan blue staining of banana leaves.

## Data Availability

Data are contained within the article or Appendix A.

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
