# Peer review of "The Autophagy-Related Musa acuminata Protein MaATG8F Interacts with MaATG4B, Regulating Banana Disease Resistance to Fusarium oxysporum f. sp. cubense Tropical Race 4"

_jof, 2024, doi:10.3390/jof10020091_

Round 1

Reviewer 1 Report

Comments and Suggestions for Authors

This paper reveals the presence of ATG8 genes in banana, which are essential for autophagy initiation, and that one of them, ATG8F, may be involved in resistance to FOTR4, the causal agent of serious banana disease. The paper also reveals the possibility that ATG8F may interact with the autophagy-related protein ATG4B. This paper contains novel findings and is expected to provide useful information to JoF readers. I think the paper is suitable for publication in the SI of JoF, but small corrections are needed on the following points.

1)L6: “and” should be deleted.

2)L27: “almost all”: Should be stated in exact numbers.

3)Materials and Methods: the authors should indicate the manufacturer, location, country, or URL of all equipment, reagents, and software used in the experiment or analysis. For example, TBtools (L110), The Illumina HiSeq X Ten platform (136), pCAMBIA1300-GFP (L142), Invitrogen Y2H system (L169).

4)L284-285: Authors should state that the expression of the MaATG8 gene is in the roots.

5)Figure 6E: Please indicate nuclear and cytoplasm by arrows or something similar so that the reader can understand them. Is there a photo that shows autophagosomes.

6)L401: “AIM” may be “ATG8-family interacting motif (AIM)”.

Reviewer 2 Report

Comments and Suggestions for Authors

The Fusarium wilt of banana is an important jeopardizing the food of several hundred million people. As the grown banana varieties are very closely related, the genetic uniformity of the host plant enhances spreading of the disease. Therefore, every experimental result, idea than can help to lessen the problem is important and needs support.

Materials and Methods

In the MM I did not fine references. This means that all methods were evaluated by you. Its probability is low.

Line 99. Why these varieties were selected for the experiment and what is their genetic background.  I suppose, it was not an accident that these genotypes were chosen. It would also be nice to know, what was the role of Nicotiana benthamiana  in the test.  

Line 118. I think, M acuminata  and its gene  belonging to the MaATG8 gene family was known about its anti-Fusarium effect. Please, cite the literature source. If not, why this gene was selected for the study?

Line 134. Why the Foc TR4 strain II5 was selected for the study? In a good journal the material and Methods does not mean you describe what you have done, but you have to show the background, why you planned the tests so. In this case the information was not given.

Line 158. Say something about the inoculation. Did you inject thee pathogen suspension, and where,  and how much inoculum was used for one plant,  and it would be interesting to know about the experimental design, how many plants were in a replicate, how many replicates you had and did you repeat the test as independent test to show the stability of the results. In this case also a statistical test would be needed, whether the differences were significant or not.

Figure 7. The Petri dish pictures are not informative, too small. Delete it od magnify it to full line size.

Th molecular part is not my specific field, seems to be interesting, but the supporting experimental material seems to be poorly treated.

For this reason, now I cannot support the publication, I wait the corrections of the methodical part. Then it can be decided whether the paper can be printed or not.

I meet rather often with papers where the molecular part is beautiful, but the experimental part is bad. Maybe the molecular geneticist is not expert in plant pathology, and this is a problem. All molecular studies are relevant when the experiments  with plants are well planned and perfectly made.

Round 2

Reviewer 2 Report

Comments and Suggestions for Authors

The paper improved much, it was worthe to rewrite the Materials and Merthods. I would like to draw your attention to the ITC ~ International Musa Germplasm Transit Centre that is the largest in the world. Maybe, this could be helpful for you to look for more resistance gene analogs or genes in this population. I have heard from somebody that resistance to banana Fusarium oxysporum was found, but the usefulness of these plants for the consumer is very moderate. It seems me that a transmission of a good resistance gene (for the specific, inherited mostly by single or oligo genes fus res) this can be possible and transfer this gene by crossing or other means int the present existing variety would be a perfect solution. A discussion about GM can be possible, but its possibly negative effects should be checked. It is clear that this paper will not solve the problem, but can be a step ahead on a long way.

Line 89. you write that differences between resistant and susceptible cultivars are unknown. How should we understand the sentence in Line 201, where you describe two varieties resistant and susceptible. Both can be valid but need explanation.

Line 209. How do you secure the same amount of banana leaf, or you simply pressed the leaf to make a small wound and this wound, or pressed surface with some destruction  was treated by the 10 ml dsRNA? Here every word has significance.

Line 228. The statistical evaluation is yet missing from the description, even in the figures it is added where appropriate.

Figure 7. The colony pictures are too small, they do not have information value. I would take out them the Figure 7 and I would use a higher magnification as Figure 8 to see it clearly.  In this case you could write a comment to the pictures, what are the differences in colony form and structure.

Lines 448 462. As I see the pictures, the resistance is not full, only partial. Here a remark would be useful, whether these genes provide full or partial resistance. We know that many monogenic F. oxysporum resistance genes for example in cabbage, do not secure full resistance. You can find, I hope, also not related genes for resistance, therefore I would leave open this problem. We know also from wheat rusts as that most of the resistance genes do not provide immunity,  from the about 80 stem rust resistance genes maybe 5-6 do this, the rest allow more or less disease development, and look similar to a polygenic resistance determined but not specific QTLs. Be careful with conclusions.

I have found several minor problems, and you can use comment for your general conclusions, I would suggest.

Now I suggest the paper to accept, now with minor , but important improvements.
